# The Influence of Single-Walled Carbon Nanotubes on the Dynamic Properties of Nematic Liquid Crystals in Magnetic Field

**DOI:** 10.3390/ma12244031

**Published:** 2019-12-04

**Authors:** Cristina Cirtoaje, Emil Petrescu

**Affiliations:** Department of Physics, Faculty of Applied Science, University Politehnica of Bucharest, Bucharest RO-060042, Romania; cristina_cirtoaje@yahoo.com

**Keywords:** liquid crystals, nanotubes, Freedericksz transition

## Abstract

This article aims to study the impact of carbon nanotube dispersions in liquid crystals. A theoretical model for the system’s dynamics is presented, considering the elastic continuum theory and a planar alignment of liquid crystal molecules on the nanotube’s surface. Experimental calculation of the relaxation times in the magnetic field was made for two cases: when the field was switched on (τ_on_), and when it was switched off (τ_off_). The results indicate an increase of the relaxation time by about 25% when the magnetic field was switched off, and a smaller increase (about 10%) when the field was switched on, where both were in good agreement with the theoretical values.

## 1. Introduction

After the discovery of nanomaterials and the synthesis of new nanoparticles, interest in the analysis of their physical properties increased in order to identify as many applications as possible [1,2,3,4,5,6,7,8,9,10]. Carbon-based nanoparticles, such as nanotubes, graphene, fullerene, or nanotori were analyzed both from experimental [11,12,13,14,15,16] and theoretical points of view [17,18,19,20,21,22]. Carbon nanotubes (CNTs) are the most studied due to their mechanical, electrical, optical, or magnetic properties that make them suitable for many applications in material science, nano-electronics, medicine, and other fields. The interest is proved by the many types of nanotubes synthesized at a global level for applications in energy storage, automotive parts, boat hulls, sporting goods, water filters, thin-film electronics, coatings, actuators, and electromagnetic shields. In order to reach their maximum potential, they must be aligned in so-called nanotube forests [23]. Liquid crystals (LC) are a good dispersing medium for nanotubes because when dispersed in it, CNTs align their long axis parallel to the molecular director of the host [24,25], leading to complex but organized nanostructures. Thus, a more precise study of their physical properties can be realized without random orientation of the nanoparticles presented in powders or in isotropic liquid dispersions.

On the other hand, there is a high interest in using LC-based devices for applications other than display technologies. We can mention here phase modulators [26,27,28], real-time holography devices [29], or switching polarizers [30]. Newly discovered nanoparticles, such as single-walled carbon nanotubes (SWCNTs), might be used in these technologies either to obtain faster response devices or, in other cases, to design slower response devices, but whose optical parameters can be controlled.

In this study, we present a theoretical analysis of liquid crystal composites with single-walled carbon nanotubes, pointing to the advantages or disadvantages for different applications. Experiments have been conducted to check the results, and a good agreement between the two data sets was found.

The single-walled carbon nanotubes were inserted in planar aligned liquid crystal cells for which the relaxation times in a magnetic field were calculated. The thermotropic nematic liquid crystal used in this article (MLC6602 from Merk) had positive magnetic anisotropy, so its molecular long axis was aligned parallel to the applied magnetic field. During this alignment process, the liquid crystal birefringence changed and, as consequence, a laser beam crossing through the sample presented intensity maxima and minima that were damped down after a period of time. This emergent intensity was experimentally recorded by automatic measuring every 0.2 milliseconds. The intensity minima number and their distribution in time were used to calculate a specific parameter of the cell, called the relaxation time. This parameter is related to the system’s ability to respond to the applied field. A dynamic study of a nematic liquid crystal composite with single-walled carbon nanotubes revealed an increase of the relaxation time for the applied field when compared to the one obtained for the reference LC. Similar results were obtained when the field was switched off after being applied for a long period of time, when the nanotubes were completely oriented by the previously applied field. Research performed on similar systems in Reference [31] revealed that CNT presented relatively high magnetic anisotropy, and that they oriented themselves with the long axis parallel to the applied field. A theoretical model for the dynamic behavior of this composite in magnetic field is proposed, based on the elastic continuum theory. The results were compared to experimental data, leading to a good agreement between them if a strong anchoring of liquid crystal molecules on CNT’s surface was considered. Thus, when the field is on, the molecule’s reorientation is “blocked” by the anchoring forces on nanotubes, which aligns much harder than the molecules as they have a large mass. The same principle works when the field is switched off after being applied on the sample for a long period of time.

## 2. Molecular Interaction with the Inserted Carbon Nanotubes and Critical Field for Freedericksz Transition

According to the model presented by Burylov and Zakhlevnykh in [32], the free energy density caused by the interaction between the nanotube and liquid crystal molecule is:(1)$$F_{nn}=\frac{wL}{2R}\left(1-3\cos^{2}\alpha \right){\left(\vec{u}\cdot \vec{n}\right)}^{2}$$
where w is the interaction energy of the LC molecule with single walled carbon nanotubes (SWCNT) surface, L is the nanotube’s length, R is its radius, $\vec{u}$ is the unit vector of nanotube’s axis, $\vec{n}$ is the nematic molecule orientation vector, α is the angle made by this vector with the nanotube’s surface, and f is the volumetric fraction of SWCNTs.

Considering the parallel alignment of carbon nanotubes with the molecular director (α=0) suggested by Patrick in [24], Equation (1) becomes:(2)$$F_{nn}=-\frac{wL}{R}{\left(\vec{u}\cdot \vec{n}\right)}^{2}$$

For a homogenous (planar) alignment of the LC cell, the carbon nanotube orientation is parallel to the glass support (Figure 1a). Due to their large mass and length (compared to liquid crystal molecules), the carbon nanotube reorientation under the external field is slower than that of liquid crystal molecules. Thus, we may assume that their direction is parallel to the glass support when calculating the relaxation time immediately after the field is applied, as shown in Figure 1b.

The total free energy density of the system (*F_t_*) also contains the free density of the elastic forces inside the liquid crystal and the free energy density due to the liquid crystal interaction with the magnetic field:(3)$$F_{t}=\frac{1}{2}\left(K_{1}\cos^{2}\theta +K_{3}\sin^{2}\theta \right)\theta_{z}^{2}-\frac{1}{2}\mu_{0}^{-1}\chi_{a}B^{2}\sin^{2}\theta -\frac{wf}{R}{\left(\vec{u}\cdot \vec{n}\right)}^{2}$$
where $\theta_{z}=d\theta /dz$ is the distortion angle gradient between the cell plates, *K*_1_ and *K*_3_ are the splay and bend elastic constants, $\chi_{a}$ is the liquid crystal’s magnetic anisotropy, and µ_0_ is the magnetic permeability.

Since:(4)$${\left(\vec{u}\cdot \vec{n}\right)}^{2}=\cos^{2}\theta =1-\sin^{2}\theta$$

Equation (3) becomes:(5)$$F_{t}=\frac{1}{2}\left(K_{1}\cos^{2}\theta +K_{3}\sin^{2}\theta \right)\theta_{z}^{2}-\frac{1}{2}\mu_{0}^{-1}\chi_{a}\left(B^{2}-\frac{2f\mu_{0}w}{\chi_{a}R}\right)\sin^{2}\theta$$

By applying the Euler Lagrange equations and using the same procedure as used in [33,34], we obtained the critical field for magnetic Freedericksz transition in the liquid crystal composite with carbon nanotubes:(6)$$\tilde{B_{c}^{2}}=B_{C}^{2}+\frac{2f\mu_{0}w}{\chi_{a}R}$$
where $B_{c}$ is the magnetic Freedricksz transition threshold for liquid crystal:(7)$$B_{c}=\frac{\pi }{d}\sqrt{\frac{\mu_{0}K_{1}}{\chi_{a}}}$$

## 3. Dynamic Behavior of Nematic Liquid Crystal with the Insertiona of Carbon Nanotubes

For the dynamic study of the mixture, the deviation angle time dependence affects the total free energy density through the dissipative term $-\frac{1}{2}\gamma {\left(\frac{\partial \theta }{\partial t}\right)}^{2}$, as shown in [35,36]. Thus, the free energy density becomes:(8)$$F_{t}=\frac{1}{2}\left(K_{1}\cos^{2}\theta +K_{3}\sin^{2}\theta \right)\theta_{z}^{2}-\frac{1}{2}\mu_{0}^{-1}\chi_{a}\left(B^{2}-\frac{2f\mu_{0}w}{\chi_{a}R}\right)\sin^{2}\theta -\frac{1}{2}\gamma {\left(\frac{\partial \theta }{\partial t}\right)}^{2}$$
where γ is the liquid crystal’s rotational viscosity coefficient.

The carbon nanotubes are long and heavy, and their orientation is practically unchanged when the field is applied. So, we can assume their long axes remain parallel to the molecular director of undisturbed nematic ${\vec{n}}_{0}$. The Euler–Lagrange equation for the deviation angle, θ leads to:(9)$$\left(K_{1}\cos^{2}\theta +K_{3}\sin^{2}\theta \right)\theta_{zz}+\left(K_{3}-K_{1}\right)\sin\theta \cos{\theta \theta }_{z}^{2}+\mu_{0}^{-1}\chi_{a}\left(B^{2}-\frac{2f\mu_{0}w}{\chi_{a}R}\right)\sin\theta \cos\theta =\gamma \frac{\partial \theta }{\partial t}$$
where we denoted the second derivative of the deviation angle versus the z coordinate across the cell by $\theta_{zz}=\frac{\partial^{2}\theta }{\partial z^{2}}$.

For small deviations, Equation (9) becomes:(10)$$\left(K_{1}+K_{3}\theta^{2}\right)\theta_{zz}+\left(K_{3}-K_{1}\right){\theta \theta }_{z}^{2}+\mu_{0}^{-1}\chi_{a}\left(B^{2}-\frac{2f\mu_{0}w}{\chi_{a}R}\right)\theta =\gamma \frac{\partial \theta }{\partial t}$$

Considering a strong anchoring of nematic molecules on the cell’s support, the deviation angle θ(z,t) varies from 0 on the edge of the cell to the maximum value (θ_*m*_) in the middle of the cell, so it can be written as:(11)$$\theta =\theta_{m}\left(t\right)\cos\frac{\pi z}{d};\;z\in \%\left[-\frac{d}{2},\frac{d}{2}\right]$$

Thus, the time dependency of the maximum deviation θ*_m_* is given by:(12)$$\left(\frac{B^{2}}{\tilde{B_{c}^{2}}}-1\right)\theta_{m}-\left[\frac{B^{2}}{B_{c}^{2}}-1+\frac{\mu_{0}K_{3}}{\chi_{a}}{\left(\frac{\pi }{d}\right)}^{2}\right]\frac{\theta_{m}^{3}}{3}=\frac{\mu_{0}\gamma }{\chi_{a}\tilde{B_{c}^{2}}}\left(\frac{d\theta_{m}}{dt}\right)$$

After solving Equation (12), we obtain:(13)$$\theta_{m}^{2}\left(t\right)=\frac{\theta_{m}^{2}\left(\infty \right)}{1+\left[\frac{\theta_{m}^{2}\left(\infty \right)}{x_{0}}-1\right]exp\left(-\frac{t}{\tau_{A}}\right)}$$
where $\theta_{m}\left(\infty \right)$ is the maximum deviation angle for an infinite time after the field was applied, $x_{0}$ is a parameter depending on the cell’s initial conditions, and $\tau_{A}$ is the relaxation time of the system when the field is switched on:(14)$$\tau_{A}=\frac{\gamma }{2\mu_{0}^{-1}\chi_{a}\left(B^{2}-\tilde{B_{c}^{2}}\right)}$$

When the field is switched off (Figure 2), after being applied for a long period of time, the carbon nanotubes remain aligned with the field direction and slow the molecular relaxation back to planar alignment. In this case, the ${\left(\vec{u}\cdot \vec{n}\right)}^{2}$ term from Equation (1) becomes:$${\left(\vec{u}\cdot \vec{n}\right)}^{2}=\sin^{2}\theta$$
and considering B=0 in Equation (8), the total free energy density is:(15)$$F_{t}=\frac{1}{2}\left(K_{1}\cos^{2}\theta +K_{3}\sin^{2}\theta \right)\theta_{z}^{2}-\frac{fw}{R}\sin^{2}\theta -\frac{1}{2}\gamma {\left(\frac{\partial \theta }{\partial t}\right)}^{2}$$

The Euler–Lagrange equation for small deviation angles is:(16)$$\left(K_{1}+K_{3}\theta^{2}\right)\theta_{zz}+\left(K_{3}-K_{1}\right)\theta_{z}^{2}-2\frac{fw}{R}\left(\theta -\frac{2\theta^{3}}{3}\right)=\gamma \frac{\partial \theta }{\partial t}$$
with solution:(17)$$\theta_{m}^{2}\left(t\right)=\frac{\theta_{m}^{2}\left(0\right)}{{\alpha \theta }_{m}^{2}\left(0\right)+\left[1-{a\theta }_{m}^{2}\left(0\right)\right]\exp\left(\frac{t}{\tau_{B}}\right)}$$
where a is a parameter depending on elastic properties and interaction energy. The relaxation time when the field is off $\left(\tau_{B}\right)$ is:(18)$$\tau_{B}=\frac{\gamma d^{2}}{2\pi^{2}K_{1}\left[1-\frac{2wf}{K_{1}R}{\left(\frac{d}{\pi }\right)}^{2}\right]}$$

As can be seen, the relaxation time when the field is off does not depend on the previously applied field.

Liquid crystals are optical anisotropic media, so the light intensity crossing through them is given by:(19)$$I=I_{0}\sin^{2}\frac{\Delta \varphi }{2}$$
where Δφ is the phase difference between ordinary and extraordinary rays,
(20)$$\Delta \varphi =\frac{2\pi }{\lambda }\int_{-d/2}^{d/2}\left(n_{eff}-n_{0}\right)dz$$
and
(21)$$\frac{1}{n_{eff}^{2}}=\frac{\sin^{2}\theta }{n_{0}^{2}}+\frac{\cos^{2}\theta }{n_{e}^{2}}$$

Due to the molecular director reorientation induced by the applied field, the difference between the ordinary and extraordinary rays also changes and the emergent beam has a series of maxima and minima. The minima correspond to an integer multiple of 2π for Δφ. Considering the distortion angle given by Equation (11), we can express the intensity minima number versus time as:(22)$$N_{m}\left(t\right)=\frac{d\Delta n}{2\lambda }\left(\theta_{m}^{2}-\frac{\theta_{m}^{4}}{4}\right)$$

When the field was switched on, we started counting the minima from the first one, while when the field was switched off, we started counting from the last one [32].

The minima number can be experimentally obtained from the intensity versus the time distribution. We could calculate the relaxation times by fitting the minimum number versus the time plot with the function given in Equation (22), where the maximum deviation angle is given by Equation (13) if the field is on, and by Equation (17) if the field is off.

## 4. Set-Up and Procedure

Experimental set-up presented in Figure 3 consists of a Weiss electromagnet which can provide a maximum magnetic field of 1 T. A 632.8 nm He-Ne laser was used to send the beam through the sample placed in the middle of the space between the electromagnet poles where a uniform intensity of the field could be obtained.

The electromagnetic poles had small holes that allowed the beam to cross through them, but did not significantly affect the magnetic field. A water cooling system was used to keep the pole’s temperature constant. A temperature rise of the electromagnetic pole must be avoided, not only because it will affect the magnetic field, but also because it would take the sample into an unstable phase at the nematic–isotropic transition border. The field calibration curve (Figure 4) was built using a Hall probe placed in the LC cell support, indicating an inner field (from the pole’s remagnetisation) of 0.0062 T and a calibration factor of 0.037 T/A.

Outside the electromagnet, there are two crossed polarizers. Their axes were oriented 45 degrees to the laser polarization axis to allow a uniform intensity distribution between the ordinary and extraordinary rays in the liquid crystal. A Thor Labs PDA 36A photovoltaic cell with a Si sensor and built-in multiplier was used for emergent intensity recording. The emergent beam intensity was recorded every 0.2 ms, and the intensity versus time plot was printed.

The LC-cell glass plates were previously prepared for homogeneous alignment using 0.1% of Polyvinyl alcohol (PVA) solution. The PVA solution was deposited on the glass plates by spin-coating and ‘’baked’’ for 1 h at 120 °. After cooling, the plates were rubbed with a soft cloth to print the grooves for planar alignment. The glass plates were connected to each other using 180 μm glass lamella as spacers. One standard sample was made by filling the cell with Merck MLC 6602 liquid crystal, and the other was filled with a mixture of the same crystal and 0.5% (volumetric fraction) of single-walled carbon nanotubes (SWCNTs) from Aldrich, having a length between 1 μm and 10 μm and a diameter of 8 nm. The concentration choice was motivated by the nanotube agglomeration we observed at higher concentrations (above 1% volumetric fraction). The same effect was reported in [37]. The microsized clusters formed at higher concentration affected the molecular orientation around their surface completely differently to well-dispersed nanotubes, so the presented model could no longer be applied.

## 5. Results and Discussions

Using the same procedure as the one presented in [36], the liquid crystal’s parameters were estimated. The values of the elastic constants obtained are $K_{1}=\left(9.1\pm 0.5\right)\times \;10^{-12}$ N, $K_{2}=\left(13.7\pm 0.5\right)\times \;10^{-12}$ N, and the rotational viscosity coefficient is γ=0.11±0.01 Pa·s. The provider indicated the liquid crystal’s birefringence to be Δn=0.0763.

For the theoretical calculation of the relaxation times, magnetic anisotropy ($\chi_{a}$) and anchoring energy density (w) were needed. They were estimated from the magnetic Freedericksz transition threshold, which was experimentally determined for each sample from the intensity versus magnetic induction plot presented in Figure 5. For this plot, a constant magnetic field was applied to the sample, and the emergent intensity was recorded. Each field was applied for 2 min, and the average emergent intensity during this period was recorded. The transition critical field corresponds to the point where the intensity presents a strong decrease. The estimated critical fields are $B_{C}=0.0835\;\pm 0.0039$ T for the standard LC sample and ${\tilde{B}}_{C}=0.0875$ ± 0.0038 T for the SWCNT + LC composite. They are close to each other at the outer limit of the error bars, so the difference between them may be considered. As it can be noticed, the critical fields are both much higher than the remnant field in the electromagnet poles, so the remagnetisation process did not affect the measurements. From Equations (6) and (7), we obtained the crystal magnetic anisotropy $\chi_{a}=\left(5.0\pm 0.3\right)\times 10^{-7}$ and the anchoring energy $w=\left(1.1\pm 0.1\right)\times 10^{-9}$ N·m to be used in the calculation of relaxation times.

For the dynamic measurements of the relaxation times, a magnetic field higher than the critical Freedericksz transition threshold was suddenly applied, the intensity versus time plot was recorded, and the minima were counted (Figure 6).

The minima number versus time was plotted and fitted with Equation (22) using Equation (13) for the deviation angle $\theta_{m}$ (Figure 7). The parameters of the fitting curve were used to calculate the experimental relaxation time. The theoretical one was determined with Equation (14).

The results for experimental and theoretical relaxation times of both samples are presented in Table 1. As it can be noticed, the experimental and theoretical results are in good agreement, especially for lower magnetic fields where the small angle approximation fits the experimental conditions.

As expected, the relaxation times for reference LC are smaller than that of the SWCNT-containing sample because the nanotubes are heavier than liquid crystal molecules and not so rapidly oriented by the applied field. Thus, they remain horizontal, and the surrounding molecules’ orientation with the applied field is dampened by the anchoring forces on CNT’s surface.

The used carbon nanotubes present positive magnetic anisotropy, as shown in [31], and after a long period of time since the field was applied, they change their orientation by aligning their long axis parallel to the field.

The magnetic was switched off one hour after it was applied on the sample, so the nanotubes were assumed to be oriented with the field. The intensity versus time plot was recorded (Figure 8), and the minima number versus time was plotted and fitted with the formula from Equation (22) using Equation (17) for the deviation angle $\theta_{m}$ (Figure 9). The theoretical relaxation time was determined with Equation (18).

Similar calculations were performed for other different magnetic fields, and the results are presented in Table 2.

Due to their large mass compared to nematic molecules, the nanotubes can be seen to present high inertia, and their alignment with the applied field is slower despite their large magnetic anisotropy. Nematic molecules, on the other hand, are very small and their alignment to the field is easier. So, when the field is switched on, the carbon nanotube remains in its original position, and the nematic molecule tends to align with the field. When the field is applied for a long period of time, the carbon nanotubes are in a vertical position (aligned to the magnetic field), and when the field is switched off, the molecules come back to planar alignment, while the carbon nanotubes remain vertical due to their high inertia. In both cases, the movement is slower than the one of pure crystal molecules, as the longer relaxation time indicates. Thus, the molecular director rotation beyond the equilibrium position settled by the applied field (the backflow effect) is also reduced.

When the field is high enough to overcome the nanotube’s inertia, the molecular dynamics change and the differences between the experimental and theoretical values increase, as seen in the last values of Table 1 and Table 2.

It can also be observed from Table 1 that the addition of SWCNTs increases the relaxation time for the applied field by about 10%, while when the field is switched off (Table 2), this value presents an average increase of 25%. This can also be explained by the positive magnetic anisotropy of the nanotubes, which helps the nanotube to rise in a vertical position. When the field is switched off, the elastic forces acting between the molecules are too weak to withdraw the nanotubes in their original position as fast as they would with the molecules, so the relaxation is much slower.

## 6. Conclusions

This manuscript presented a theoretical analysis of the effects of single carbon nanotube insertion in liquid crystal cells, pointing to both the advantages and disadvantages of these composites. The model was based on the elastic continuum theory and the anchoring forces of liquid crystal molecules on CNT’s surface. As it could be observed, the Freedericksz transition threshold and the relaxation times were increased. The increase of the Freedericksz transition threshold meant that a stronger field was needed to distort the molecule and hence a higher power consumption, so adding carbon nanotubes to a liquid crystal cell would clearly be a disadvantage for display devices. Longer relaxation times are also an undesirable effect for any display, as nobody would want to increase the response time. The increase of the relaxation times may be an advantage only for a phase modulation system, where a slow variation of the molecular deviation angle can be used to control the effective refractive index.

## Figures and Tables

**Figure 1 materials-12-04031-f001:**
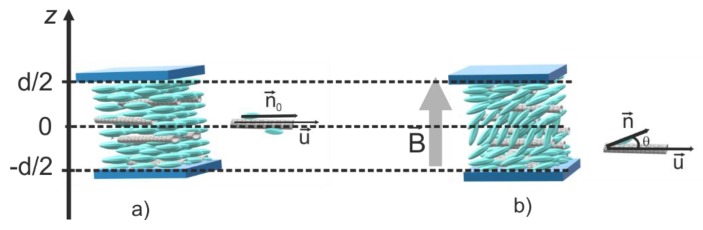
Carbon nanotube alignment in homogeneous (planar) aligned cells subjected to a magnetic field $\vec{B}$: (**a**) nanotube and liquid crystal molecule orientation before the field was applied; (**b**) nanotube and liquid crystal molecule orientation when the field was applied.

**Figure 2 materials-12-04031-f002:**
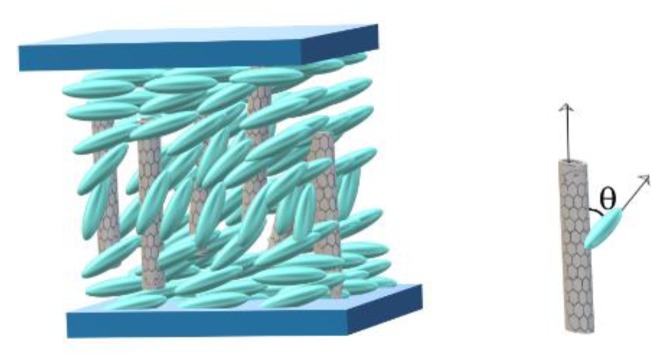
Carbon nanotube orientation a long time after the magnetic field was applied onto the sample. When the field is switched off, the nanotube remains vertical, while the molecules tend to go back to planar alignment being attracted by the elastic intermolecular forces and surface anchoring.

**Figure 3 materials-12-04031-f003:**
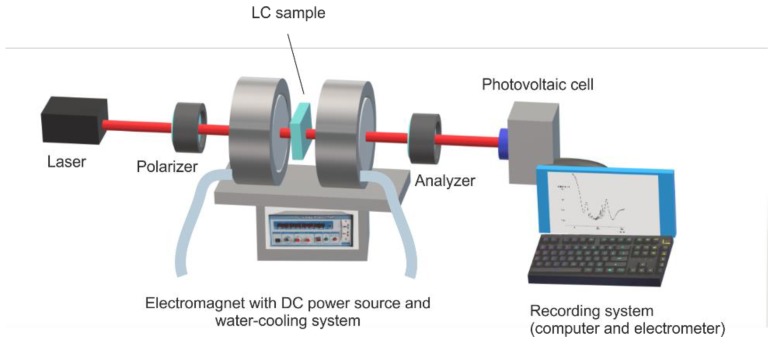
Experimental set-up for the dynamic measurement of emergent laser beam intensity through the sample subjected to a magnetic field.

**Figure 4 materials-12-04031-f004:**
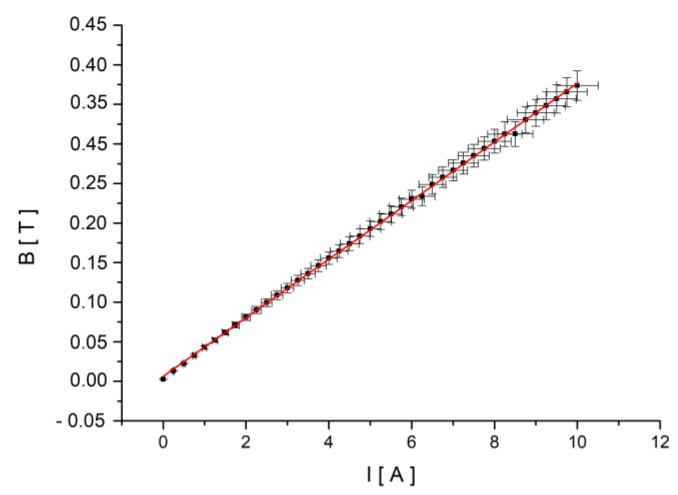
Calibration curve. The current through the electromagnet coils was increased by 0.25 A for each measurement. Each point represents the average value of ten recorded values of the induced magnetic field (B) at the midpoint of the distance between the poles (the sample slot).

**Figure 5 materials-12-04031-f005:**
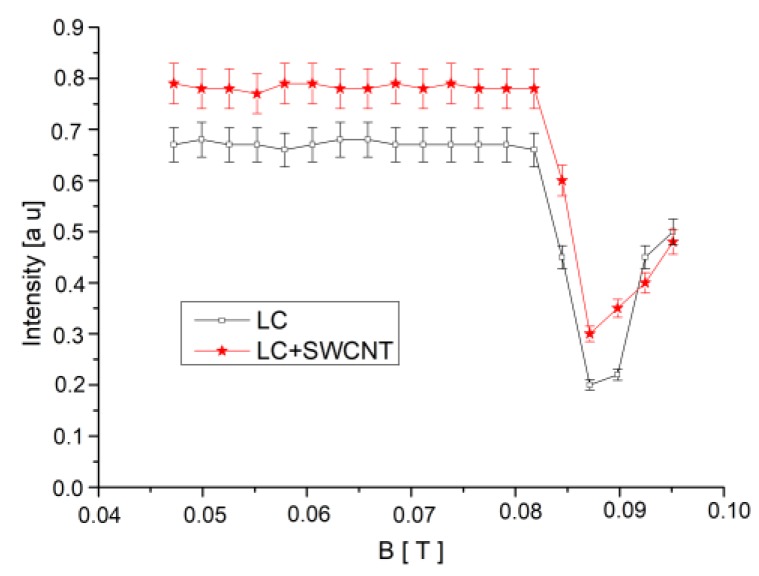
Laser beam intensity versus magnetic induction plot and Freedericksz transition threshold evaluation. Lines are guides for the eyes.

**Figure 6 materials-12-04031-f006:**
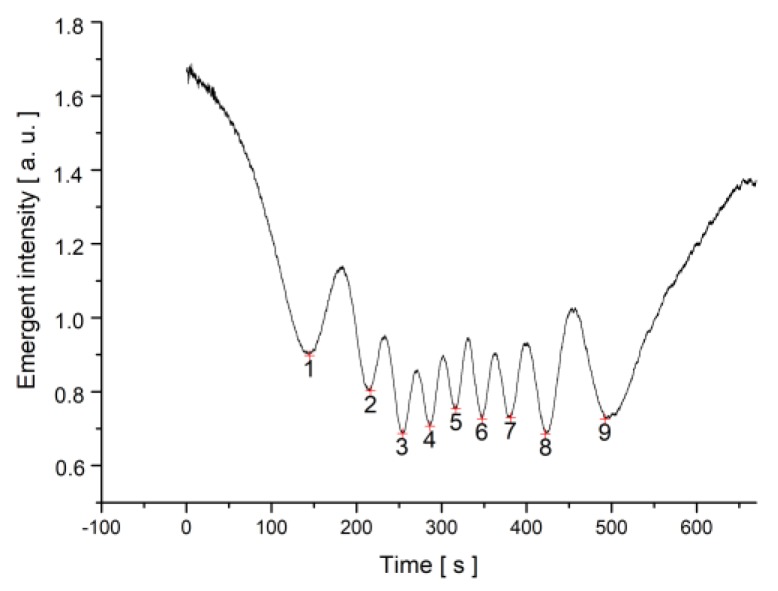
Laser beam intensity versus time plot when a magnetic field of B = 0.1074 T was applied to the SWCNT + MLC6602 mixture sample.

**Figure 7 materials-12-04031-f007:**
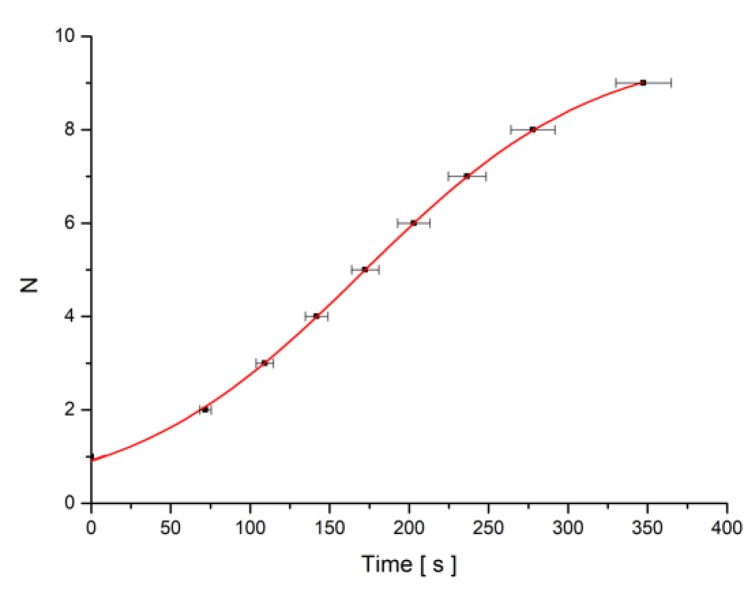
Minima number (N) versus time plot for SWCNT + MLC6602 mixture sample when the field B = 0.1074 T is switched on.

**Figure 8 materials-12-04031-f008:**
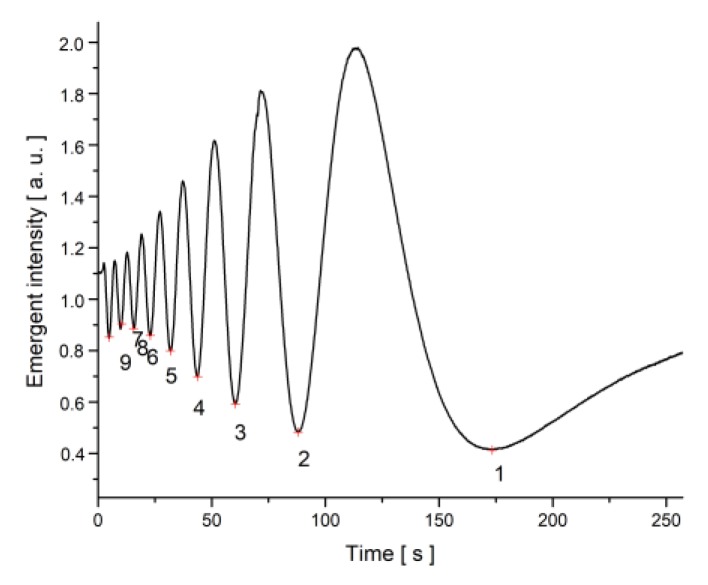
Intensity versus time plot for SWCNT + MLC6602 when a magnetic field B = 0.1114 T was switched off.

**Figure 9 materials-12-04031-f009:**
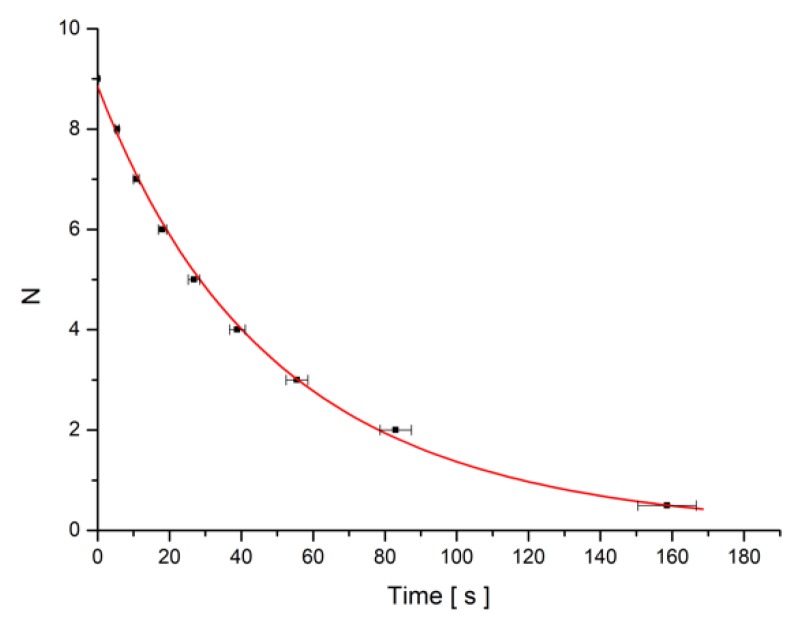
Minima number versus time plot SWCNT + MLC6602 when a magnetic field B = 0.1114 T was switched off.

**Table 1 materials-12-04031-t001:** Relaxation times for the standard sample containing Merck 6602 and for SWCNTs + Merck 6602 samples when the magnetic field was switched on.

B(T)	LC		LC+SWCNT	
	$\tau_{A}\;\mathrm{Exp}\;(s)\;$	$\tau_{A}\;\mathrm{Theor}\;(s)$	$\tau_{A}\;\mathrm{Exp}\;(s)$	$\tau_{A}\;\mathrm{Theor}\;(s)$
0.1035 ± 0.00517	33.5 ± 1.8	36.9 ± 1.9	37.6 ± 1.9	39.5 ± 1.9
0.1074 ± 0.00537	28.6 ± 1.7	30.2 ± 1.5	31.5 ± 1.6	32.0 ± 1.6
0.1114 ± 0.00557	25.7 ± 1.7	25.4 ± 1.3	28.9 ± 1.5	26.6 ± 1.3
0.1153 ± 0.00577	25.2 ± 1.6	21.8 ± 1.1	26.0 ± 1.3	22.7 ± 1.4

**Table 2 materials-12-04031-t002:** Relaxation times when the field was switched off for the standard sample containing Merck 6602, and for SWCNTs + Merck 6602 samples.

B(T)	LC		LC+SWCNT	
	$\tau_{B}\;\mathrm{Exp}\;(s)$	$\tau_{B}\;\mathrm{Theor}\;(s)$	$\tau_{B}\mathrm{Exp}\;(s)$	$\tau_{B}\;$ **Theor (s)**
0.1035 ± 0.00517	63.3 ± 3.2	62.3 ± 3.1	80.0 ± 4.0	78.2 ± 3.9
0.1074 ± 0.00537	64.1 ± 3.2		81.3 ± 4.1	
0.1114 ± 0.00557	62.9 ± 3.1		80.0 ± 4.0	
0.1153 ± 0.00577	63.7 ± 3.2		80.4 ± 4.0	

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
