# Peer review of "The Influence of Single-Walled Carbon Nanotubes on the Dynamic Properties of Nematic Liquid Crystals in Magnetic Field"

_materials, 2019, doi:10.3390/ma12244031_

Round 1

Reviewer 1 Report

Manuscript ID: Materials-647957

Title: The Influence of Carbon Nanotubes on The Dynamic Properties of Nematic Liquid Crystals in Magnetic Field

This paper presents a theoretical model to study the impact of carbon nanotubes (CNTs) on the dynamic behavior of nematic liquid crystals. The author showed that by embedding CNTs into the liquid crystal matrix, a precise control of the nematic director orientation can be obtained applying an external magnetic field. I believe that the topic is timely and important and the presented theoretical model is quite novel. In my opinion, the work can be published in Materials considering the following corrections needed in the manuscript or points that require more clarification and explanations:

The conclusion part needs to be revised as it is evidently does not reflect the concept and the key results of the work. The introduction part needs to be revised as it does not reflect the motivation of the author to establish its approach. There are some grammatical errors and typos in the text (not an exhaustive list, authors need to re-check the whole document for spelling).

Author Response

We want to thank to editor and to all reviewers for their help and valuable observations.  Now we think we managed to increase the manuscript’s quality. Our answer to reviewer 1 is presented below:

We revised the Conclusion section and made some addings for the Results and Discussion section for a better connection between results and conclusions.

We also revised the Introduction section to show our motivation for the subject choice.

The language was revised, and we hope not to miss any mistakes.

Respectfully yours,

Prof. E. Petrescu

Reviewer 2 Report

In this manuscript, Cirtoaje and Petrescu study the dynamics of nematic liquid crystals doped with carbon nanotubes when a magnetic field is either applied or removed. In the studies, the authors find that the magnetic response (Frederiks threshold) of the NLC is not enhanced by the presence of the CNT, and only the relaxation time when the field is switched off is significantly increased by the presence of the inclusions. If I understand correctly, the authors attribute this behavior to inertial effects (“high mass of nanotubes”), which I find hard to believe.

The authors derive a model for the relaxation time, adapting results from the literature, in order to include the effect of the CNT. Data analysis seems to provide a good agreement with the predictions of this model.

Although the topic is of potential interest to the readers of this journal, the provided data does not allow to reach any definite conclusion as to the role of the CNT in the system. The effect is small, probably often within the error bars (that are not reported!). Moreover, the conclusion is confusing. If the presence of CNT increases the response time of the NLC under a magnetic field, as the data show, how can the authors conclude that “… Yet there is a bid (sic) advantage for LC-based phase modulators because the backflow effect is considerably REDUCED ….”. Moreover, I could see an advantage if the response times were reduced, but not if they are increased. This discussion needs to be clarified.

In summary, I cannot recommend acceptance of the manuscript in the current form. More experiments would need to be performed before meaningful conclusions could be reached.

In particular:

Experiments at different concentrations of CNT (f). Since both Chi_a and w should be independent on f, different experiment should yield similar estimates for those parameters. Moreover, the predicted scaling of the relaxation times with “f” could be tested. Experiments at different (smaller) cell thickness. If the authors hypothesis is true, then the effect of CNT should be relatively stronger at shorter relaxation times, and the support for the conclusions would be meaningful.

Besides these major aspects, there are other issues that the authors need to consider to improve on the clarity of the manuscript:

None of the experimental results are given with their corresponding error bars. These are important, in particular when the reported effects are weak. English grammar should be revised throughout the manuscript. Although existing mistakes do not, in general, affect the readability of the paper, they should be corrected. In the introduction, authors claim that “Thermotropic NLC have positive magnetic anisotropy”. Actually, not all of them. For instance, CCN37 (and others in the same family) has a negative magnetic anisotropy. There are sentences such as “which align much harder than the molecules” (line 40) that are too casual and imprecise. In the expression (u n) that involves vectors u and n (eq. 1 and following), I understand that the authors mean the scalar product of these two vectors. Then, the “dot” symbol is missing.

In eq. 3 (and following), the symbol “f” is used to represent BOTH the elastic energy density and the volumetric fraction of CNT. Similarly, in equation 17, “alpha” is used as a “parameter”, but in equation (1) it was defined as the anchoring angle of NLC on the CNT. This is not acceptable. Each symbol must have a single, clear meaning.

Author Response

We want to thank to editor and to all reviewers for their help and valuable observations.  Now we think we managed to increase the manuscript’s quality. Our answer to reviewer 2 is presented below:

The title format was changed, and the content was reduced to SWCNT.

The Abstract was revised focusing on the work instead of presenting the conclusions in advanced.

The figure captions and equations descriptions were revised, and we hope not to miss any more errors.

The language was revised for a clear and correct presentation. We hope we achieved this.

Equations were revised, changes were made to eliminate confusing phrases. We hope we didn’t miss any of them.

Cited figures, equations and references were also checked and presented as you suggested for a better presentation.

In the Experimental set-up section, we made some changes to clarify the work conditions:

The remagnetisation process does not affect the measurements because the feromagnetic field is below the Freedericksz transition threshold. A Hall probe was used to plot a calibration plot. The plot was introduced in the manuscript text A water-cooling system was used to keep the temperature constant The calibration curve was added in the manuscript. The graph presented in the text is only to show present the method. We considered more useful to give the results (the values in Table 1 and Table 2) instead of printing all experimental plots. We have the plots for each magnetic field value for liquid crystal and liquid crystal+SWCNT. Please let us know if you consider they should all be included in the manuscript and we can add them.

Respectfully yours,

Prof. E. Petrescu

Reviewer 3 Report

referee report
materials-647957
THE INFLUENCE OF CARBON NANOTUBES ON THE DYNAMIC PROPERTIES OF NEMATIC LIQUID CRYSTALS IN MAGNETIC FIELD
Cristina Cirtoaje and Emil Petrescu

This manuscript reports on the influence of single-walled carbon nanotubes on the orientation of a liquid crystal cell to an applied
magnetic field. The topic is interesting, and fitting to Materials.

Several general points need, however, attention:
# The English requires considerable improvement. Please have a native speaker checking the manuscript.
# Please ensure that all abbreviations used are defined in the text. Currently, "LC" is never defined in the text.
# There should always be a space between a physical quantity and its unit. Please check the entire text and graphs.
# There should always be a space between text and a reference.
# Please use proper SI units throughout the manuscript.
# The figures are mostly well prepared, but the figure captions do not properly describe the content of the figure. As there is no page
limit, please be more precise and explain the content of the figure properly.
# In the reference list, all the titles also require proper formatting of chemical formulae.

Specific remarks:

# The title should not be capitalized. Also, it should be formulated more precisely -- the work is about SWCNTs, not general.
# The abstract should be more focused on the work presented, and not give the conclusions in advance.
# The equations and their descriptions should be written with more care: Often, not all elements are properly explained. Sometimes,
an expression like "term 1" appears which should be properly "equation 1" (otherwise one would understand "term 1 in the previous eq." --
which is not meant here). Make sure that all elements in the equations are properly defined.
# When citing a figure in the text, please use brackets (otherwise the text gets very messy), or mention the figure in the text like
"Figure 1 presents...", "In Fig. 1, there are...".
# The electromagnet uses ferromagnetic pole pieces to generate the fields. Does the remagnetization process affect the observations?
# Fig. 5: What is the meaning of "magnetic field of T=0.1074 T"?
# How were the magnetic fields chosen? It looks like the current through the coil was controlled, and then the field was determined via
a calibration factor. Did the authors use a Hall probe? What about warming effects?

# Figs. 7 and 8 only show the behavior of the LC sample. Why there is no graph for the LC+SWCNT sample? The caption of Fig. 7 gives a wrong
field value (compare with table 2). Why only only one field was measured?

Overall, the topic would be interesting, but the experimental part of the manuscript contains many faults and problems.
Therefore, the present manuscript is not suitable for publication. For a resubmission, considerable work is required
to obtain reasonable results, prepare proper figures, in a proper style and a reasonable discussion of the results obtained.

Author Response

We want to thank to editor and to all reviewers for their help and valuable observations.  Now we think we managed to increase the manuscript’s quality. Our answer to reviewer 3 is presented below:

The idea of considering the inertial effect in discussing the SWCNT behavior in liquid crystal appeared not only from this experiment but also in reference [18] where we discussed similar systems in electric field but in slightly different conditions: the voltage was switched off immediately after recording the dynamic response for applied field. Since the relaxation times in electric fields are very small, we wanted to check the results in magnetic field where these parameters are larger and expect the measurement errors to be smaller. The effect on Freedericksz transition is small indeed and it might be neglected for some practical applications but for the relaxation times there are considerable differences varying from 10% to 25% of reference data. Your are perfectly right about the benefits of reducing the response time when SWCNTs are added in liquid crystal cell, but unfortunately this didn’t happened. In this manuscript we tried to find the cause of this undesirable effect and suggest other possible applications of the systems. For instance, in phase modulators where the simple formulas for small deviation angle may be applied for higher magnetic fields. We know from experience how hard is to work with weak magnetic fields around electronic devices and how large the experimental errors can be. We made additional explanations in Results and discussions section and we hope to have clarified the discussion. The error bars were added to the plots where it was possible. We could not do it for intensity versus time plot because there are too many points recorded and the intensity minima would be harder to view. The whole manuscript was revised for English corrections and we hope to have corrected all the spelling or grammar mistakes. About the magnetic anisotropy we clarified the phrase. The liquid crystals used by us have positive anisotropy. We corrected it in the manuscript The equations were corrected at your suggestions and no more identical notations were used for different parameters. About your suggestion about testing the model for different SWCNTs concentrations, we must tell you that we tried to do it. Here are our observations: If the concentration is 1% or higher, clustering effects occurs, and the nanotubes are gathering together in large microscopic bulks that could be observed in optical microscope. For lower concentrations, it is quite hard to determine precise variations.

Respectfully yours,

Prof. E. Petrescu

Round 2

Reviewer 2 Report

The authors have taken some steps to improve the manuscript, but they have not fully corrected some issues nor properly discussed some of my criticisms. Therefore, before I can recommend acceptance, these aspects must be corrected:

English has improved, but it is not satisfactory yet. Some math errors have not been corrected. For instance, the “dot product” symbol is missing in several places. After eq. 16 you have written eq. 19. It should be “17” Error estimation should be given for ALL computed parameters, so that they can be properly compared. This includes values in table 1, 2, and other parameters listed in the text. In the updated conclusions, the authors state that “Due to the nanotubes inertia, one can use higher magnetic fields to increase the distortion angle with reduced backflow”. I do not understand the logics in this sentence. Concerning my criticisms regarding the need for more experiments, the authors argued that the concentration of CNT could not be changed because concentrations higher than 1% led to CNT aggregation, and lower concentration led to even smaller effects. This is something you should discuss in the text, because it is relevant. On the other hand, I believe there is some room between 0.5% and 1% to do new experiments. The paper is weak without additional measurements. In the same line as my criticisms in point 6, and something the authors did not discuss in your reply, you could have tested thinner cells to check the behavior for faster dynamics. I am still not convinced about the interpretation of your results, since it is harder to believe that inertial effects are relevant is such long relaxation times. Experiments with thinner cells are needed.

Author Response

Dear reviewer,

Thank your for reading the manuscript so carefully. We really appreciate this whatever you decide regarding this article. Here is what we did and what we could not do from your recommendations:

We revised the language (we hope it is good now) and corrected the remained equations errors. Thank you very much for pointing them to us.

Error estimations were given for all computed and measured parameters including the data in Table 1 and Table 2.  

What I meant with that phrase was that for higher magnetic field, its standard deviation is higher than environmental magnetic field coming from electronic devices so their influence on phase modulation are reduced. We replaced the phrase to avoid confusion.

We discussed the concentration choice in the text as you suggested.

About your doubt about the effect of nanotube inertia. We must tell that this what we found to explain why the relaxation time is increased since its magnetic anisotropy is higher than those of liquid crystal.

Maybe you are right and there are other effects besides inertia that influence the relaxation time but it we ‘ll have to see it in the future. What we know so far is that it acts the same way in electric and magnetic field, and it is not the given by the molecule’s anchoring forces on CNT’s surface because they are too small.

The experimental data set must fulfill some conditions to fit in the approximations used for the theoretical model. We needed small deviation angles, good dispersion and enough intensity minima to fit the points from Fig.7 and Fig.9.

As mentioned, we observed clustering effect at 1% CNT concentration in liquid crystal, but others (reference [37]) observed it at lower concentrations. This reduce the concentration range that can be used between 0.5% and 0.6-0.7% and the errors would be high because we had to use very small amounts of substance.

We tried to use thinner cells (80 micrometers), but the number of intensity minima was too small and the plot given in fig. 7  was linear. This led to a wide range of fitting parameters and inconclusive results for the relaxation time.

What we propose here is a starting point of the theory with experimental conditions in agreement with theoretical approximations and results that seems to fit with the calculated ones. Other systems will be considered in our future work with other classes of liquid crystals and carbon nanotubes to complete this model and finally to obtain a theory that can fit for a wide range of mixtures and cells.

Respectfully yours,

Prof. Emil Petrescu

Reviewer 3 Report

The changes made to the manuscript have clearly improved the quality. Therefore, the manuscript is now suitable for publication.

Author Response

Thank you for appreciation and for helping us to improve the manuscript!

With gratitude,

Emil Petrescu

Round 3

Reviewer 2 Report

The authors have addressed my criticisms and made some required corrections. As the authors acknowledge, this work has to be considered preliminary, and I think this is now clear in the discussion and in the conclusions. 

Therefore, I recommend acceptance in the current form.